# DISTILLING VIDEO DATASETS INTO IMAGES

## ABSTRACT

Dataset distillation aims to synthesize compact yet informative datasets that allow models trained on them to achieve performance comparable to training on the full dataset. While this approach has shown promising results for image data, extending dataset distillation methods to video data has proven challenging and often leads to suboptimal performance. In this work, we first identify the core challenge in video set distillation as the substantial increase in learnable parameters introduced by the temporal dimension of video, which complicates optimization and hinders convergence. To address this issue, we observe that a single frame is often sufficient to capture the discriminative semantics of a video. Leveraging this insight, we propose **S**ingle-**F**rame **V**ideo set **D**istillation (SFVD), a framework that distills videos into highly informative frames for each class. Our method focuses on distilling videos into highly informative frames for each class for effective optimization during distillation, a framework that distills videos into highly informative frames for each class. Using differentiable interpolation, these frames are transformed into video sequences and matched with the original dataset, while updates are restricted to the frames themselves for improved optimization efficiency. To further incorporate temporal information, the distilled frames are combined with sampled real videos from real videos during the matching process through a temporal reshaping network. Extensive experiments on multiple benchmarks demonstrate that SFVD substantially outperforms prior methods, achieving improvements of up to 5.3% on MiniUCF, thereby offering a more effective solution for video dataset distillation.

## 1 INTRODUCTION

Dataset distillation (DD) (Wang et al., 2018; Zhao et al., 2020; Zhao & Bilen, 2021; Cazenavette et al., 2022; Guo et al., 2023; Cui et al., 2023; Du et al., 2023; Zhao & Bilen, 2023; Wang et al., 2022; Shang et al., 2023; Yin et al., 2023; Su et al., 2024; Wang et al., 2025; Zhao et al., 2025) aims to distill a large dataset into a condensed informative synthetic dataset, so that the model trained on it could have a similar performance as the original dataset. This small, information-rich dataset drastically reduces training time and computational costs, significantly lowers storage requirements, enables rapid prototyping and hyperparameter tuning (Poyser & Breckon, 2024), and can facilitate easier data analysis and potentially even privacy-preserving data sharing (Yang et al., 2024). While recent works (Cui et al., 2023; Guo et al., 2023; Su et al., 2024; Gu et al., 2024) have made significant progress on image dataset distillation tasks, the application of dataset distillation for video datasets remains relatively underexplored.

Videos extend images into the temporal domain, capturing not only spatial information in each frame but also motion and temporal dynamics over time. Furthermore, video recognition plays a crucial role in enabling machine learning models to understand the world, as it allows them to capture not only static appearances but also the evolving temporal patterns that characterize real-world phenomena. Recently, several studies have investigated video dataset distillation (Wang et al., 2024), but current approaches remain primarily empirical and still exhibit suboptimal performance.

In this paper, we argue that a fundamental challenge hindering effective video dataset distillation arises from the substantial increase in learnable parameters within synthetic datasets, which is caused by the added temporal dimensionality of video data compared to images. As illustrated in Figure 1(a), in image dataset distillation, each synthetic image is represented as a learnable tensor that is iteratively updated throughout the distillation process. In Figure 1(b), a straightforward extension of image distillation methods to video data results in treating each synthetic video as a fully learnable tensor.

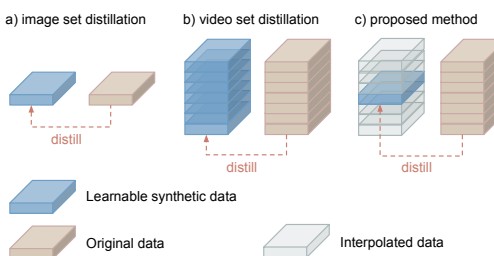
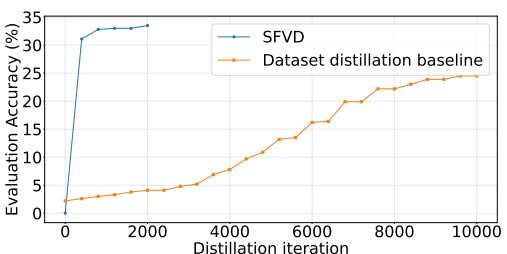

Figure 1: **Image and video dataset distillation parameter spaces.** (a) In image DD, each image in the distilled dataset is a synthesized learnable image that is updated throughout the distillation process. (b) When image distillation is directly extended to video datasets, all frames within each synthetic video are learnable. (c) Our proposed method constrains learnability to a single frame per synthetic video, thereby reducing the parameter space for effective optimization.

Figure 2: **Comparison between dataset distillation baseline and SFVD.** We implement the SOTA image dataset distillation method (Guo et al., 2023) directly on the video dataset UCF101 (Soomro et al., 2012). Orange line indicates the evaluation performance of the baseline during the distillation process. We can observe that it requires more iterations to converge and achieves suboptimal performance, whereas the proposed SFVD (blue line) converges quickly.

However, since a video inherently contains far more pixels than an image, this direct extension leads to an explosive growth of the parameter space. Such a heavy-parameter optimization makes it difficult for distillation algorithms to converge effectively (Nakkiran et al., 2021) and to comprehensively capture the complex information distribution of the original video dataset within a condensed dataset. As illustrated in Figure 2, the direct application of image dataset distillation method (Guo et al., 2023) to video dataset encounters difficulties in convergence and yields suboptimal performance.

To address the above challenge, we build directly on our preliminary observation that a single frame can capture a substantial portion of the semantic content of a video. Motivated by this finding, we propose the Single-Frame Video set Distillation (SFVD) framework. We depart from directly matching synthetic videos with the original dataset and instead propose to match synthetic images with real videos. Specifically, we represent each video by a single frame, which is subsequently interpolated into a video sequence and match with the original dataset, while only updating the parameters within the frame. This design substantially reduces the optimization burden while enabling effective video dataset distillation. As illustrated in Figure 1(c), rather than synthesizing entire video sequences, the goal of SFVD is to distill a small set of highly representative frames. This design substantially reduces the learnable parameter space during the distillation process, thereby enabling more effective and stable optimization.

To further further incorporating temporal dynamics, we integrate the distilled frames with sampled original videos as temporal cues and employ a Temporal Reshaping Network (TRN) to fuse them into a combined representation. We then apply training trajectory matching (Cazenavette et al., 2022) with the original dataset, where the distilled frames act as informative priors that guide parameter updates, ensuring robust initialization and faster convergence toward high-quality distilled video representations. We conduct extensive evaluations on four widely used video benchmarks: UCF101 (Soomro et al., 2012) and HMDB51 (Kuehne et al., 2011) for human action recognition, Kinetics (Carreira & Zisserman, 2017) for large-scale video understanding, and Something-Something V2 (Goyal et al., 2017) for fine-grained temporal reasoning. Across all datasets and evaluation settings, our method consistently surpasses existing approaches by a substantial margin. On the Mini-UCF benchmark, our approach achieves substantial improvements, outperforming state-of-the-art methods by 4.5% in absolute accuracy for IPC=1 and by up to 5.3% for IPC=5.

Our main contributions can be summarized as follows:

- We identify and analyze the significantly increased number of learnable parameters of the synthetic videos as a critical issue for video set distillation, impeding convergence and information capture.
- We propose a novel Single-Frame Video set Distillation (SFVD) framework in which a single frame can be interpolated into a video, matched with the original dataset, while only updating the parameters within the frame. Additionally, we integrate temporal information alongside the distilled frames, creating a comprehensive representation.

- We demonstrate through extensive experiments on various benchmark datasets that our proposed SFVD significantly outperforms existing methods, validating the effectiveness of our approach.

## 2 RELATED WORK

**Dataset Distillation.** Dataset distillation (Wang et al., 2018) aims to compress a large dataset into a much smaller synthetic dataset while preserving the essential information required for effective model training. It has been widely applied to tasks such as network architecture search (Elsken et al., 2019), federated learning (Li et al., 2020), and continual learning (Zenke et al., 2017). A foundational approach in dataset distillation is performance matching, where the objective is to ensure that models trained on the distilled dataset achieve comparable test performance to those trained on the original dataset. A major class of dataset distillation methods is gradient matching (Zhao et al., 2020; Zhao & Bilen, 2021), where the key idea is that models trained on synthetic data should produce similar gradients as those trained on the original dataset. Building upon gradient matching, trajectory matching (Cazenavette et al., 2022; Du et al., 2023; Guo et al., 2023) further refines this idea by aligning the model's parameter updates across training epochs, rather than just the overall gradients. Besides gradient and trajectory matching, other approaches include distribution matching (Zhao & Bilen, 2023), which focuses on aligning the feature distributions between synthetic and real datasets, and feature matching and label matching, which align feature representations and label distributions, respectively. These methods provide complementary perspectives on capturing and preserving information during dataset distillation.

Recently, to enable distillation at larger scales, decoupled optimization methods (Yin et al., 2023; Shao et al., 2024) have been proposed. These approaches separate matching into computationally efficient stages, reducing memory and time complexity. Another emerging direction leverages generative models, where a generator is trained to capture the underlying data distribution and sample synthetic instances accordingly. In addition to image datasets, there has been progress in distilling other modalities, including image-text datasets (Wu et al., 2023; Xu et al., 2024), sequential datasets (Zhang et al., 2025), medical datasets (Li et al., 2024), and video datasets (Wang et al., 2024). For video dataset distillation, the prior work (Wang et al., 2024) have mainly focused on empirical studies, but a thorough analysis of why image dataset distillation methods often fail on video data remains lacking. In this paper, we first identify the unique challenges in video dataset distillation. We then present our key observations on video model training dynamics. Based on these insights, we propose a novel method SFVD, as detailed in Section 3.

**Video Recognition.** Video classification has been extensively studied, and a wide range of architectures have been proposed to handle the spatial and temporal complexities of video data. Early approaches are based on 2D ConvNets, which is used for spatial feature extraction and a recurrent module, such as an LSTM or GRU (Donahue et al., 2015; Yue-Hei Ng et al., 2015), is stacked on top to model temporal dynamics across frames. To more effectively capture spatiotemporal features jointly, 3D ConvNets (Tran et al., 2015; Carreira & Zisserman, 2017; Feichtenhofer et al., 2019) were introduced. Hybrid architectures decompose 3D convolutions into spatial and temporal operations, as in P3D (Qiu et al., 2017), S3D (Xie et al., 2018), and R(2+1)D (Tran et al., 2018). More recently, transformer-based architectures have achieved strong performance on video tasks. Models like ViViT (Arnab et al., 2021) and Video Swin Transformer (Liu et al., 2022) adapt vision transformer structures to spatiotemporal input, while MViT (Multiscale Vision Transformer) (Fan et al., 2021) introduces a hierarchical structure for multi-scale temporal modeling. In this paper, following prior work on video dataset distillation (Wang et al., 2024), we primarily adopt C3D for evaluation, and additionally employ CNN+GRU/LSTM for cross-architecture validation.

## 3 METHODS

### 3.1 PROBLEM FORMULATION

The problem of Video Dataset Distillation is formulated as follows: given a large, real video dataset $\mathcal{T} = \{x_i, y_i\}_{i=1}^{|\mathcal{T}|}$, where $x_i \in \mathbb{R}^{f,c,h,w}$, video set distillation aims to generate a synthetic dataset $\mathcal{S} = \{\hat{x}_i, \hat{y}_i\}_{i=1}^{|\mathcal{S}|}$, where $|\mathcal{S}| \ll |\mathcal{T}|$, so that the model $\phi_\theta$ trained on synthetic dataset $\mathcal{S}$ has the similar performance on the original dataset $\mathcal{T}$. It is worthy noting that $\hat{x}_i$ is not strictly constrained to

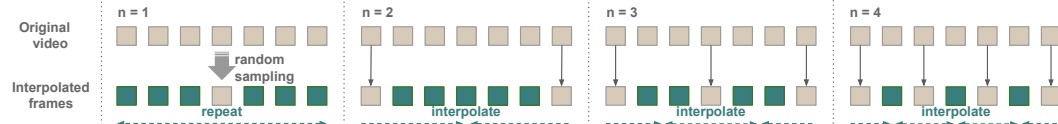

Figure 3: **Illustration of the preliminary experimental setup.** The $n$ frames selected from an original video are processed by an interpolation method $g(\cdot)$ to generate a new video sequence maintaining the original temporal length $f$. When $n = 1$, the single frame is randomly sampled. For $n \geq 2$, frames are sampled uniformly over the temporal extent of the video.

$\hat{x}_i \in \mathbb{R}^{f,c,h,w}$, provided the preprocessing $g(\cdot)$ yields $g(\hat{x}_i) \in \mathbb{R}^{f,c,h,w}$. Typical dataset distillation methods (Liu & Du, 2025) use the following formula to optimize $\mathcal{S}$:

$$\mathcal{S}^* = \arg \min_{\mathcal{S}} \; \mathbb{E}_{\theta^{(0)} \sim \Theta} [l(\mathcal{T}; \theta_{\mathcal{S}}^*)], \tag{1}$$

where $\theta_{\mathcal{S}}^* = \arg \min_{\theta} \; l(\mathcal{S}; \theta)$ is the optimal parameters trained on $\mathcal{S}$, $l(\cdot; \theta)$ represents the loss function. $\Theta$ is the initial parameter distribution. If we are using a trajectory matching strategy (Cazenavette et al., 2022), the formulation of video set distillation could be written as:

$$\mathcal{S}^* = \arg \min_{\mathcal{S}} \mathbb{E}_{\theta^{(0)} \sim \Theta} \sum_{t=0}^{T} \mathcal{D} \left( \theta_{\mathcal{S}}^{(t)}, \theta_{\mathcal{T}}^{(t)} \right), \tag{2}$$

where $\mathcal{D}(\cdot, \cdot)$ calculates the distance between the parameter trained on synthetic dataset $\theta_{\mathcal{S}}$ and original dataset $\theta_{\mathcal{T}}$ at step $t$. $T$ is the trajectory length.

### 3.2 OBSERVATION

We first reveal the primary challenge of the video set distillation against the image dataset. Then we introduce the designed preliminary experiments and illustrate how the observations from the preliminary experiment results can help with the video set distillation.

**Challenge of video dataset distillation.** The principal challenge inherent in video dataset distillation comes from the significantly increased number of learnable parameters of the synthetic videos compared to that of synthetic images in image dataset distillation. Reviewing the dataset distillation on image dataset, backpropagation operates on a synthetic dataset $\mathcal{S} = \{\hat{x}_i, \hat{y}_i\}_{i=1}^{|\mathcal{S}|}$, where $\hat{x}_i \in \mathbb{R}^{c,h,w}$. However, for video set distillation, the back propagation takes effects on a synthetic video dataset $\mathcal{S} = \{\hat{x}_i, \hat{y}_i\}_{i=1}^{|\mathcal{S}|}$, where $\hat{x}_i \in \mathbb{R}^{f,c,h,w}$, with $f$ denoting the number of frames. Because of this additional dimension $f$, the number of learnable parameters for each sample $\hat{x}_i$ increases substantially compared to image dataset distillation. This enlarged parameter space makes trajectory matching, which directly performs pixel-wise updates on the synthetic dataset, considerably more difficult to optimize and less stable. As a result, directly applying image dataset distillation methods (Cazenavette et al., 2022; Zhao & Bilen, 2023; Guo et al., 2023) often leads to convergence issues and ultimately yields suboptimal performance in the video domain, as empirically demonstrated in Figure 2.

**Frame information observation.** In light of the aforementioned challenges in video set distillation, identifying and prioritizing the distillation of the most salient information becomes crucial. The temporal dimension in video dataset distillation often contains redundant or less critical information for discriminative learning tasks. Building on previous research (Zhu et al., 2018), we first hypothesize that the core semantic content necessary for model training might be preserved even with a significantly reduced set of frames. This hypothesis motivates an investigation into the efficacy of learning from video sequences constructed by interpolating frames, as a potential avenue to mitigate the challenges posed by high-dimensional video inputs. To explore this, we designed preliminary experiments to quantify the information a model can acquire from such sparsely sampled representations.

In our experimental setup, for each original video, we sampled $n$ frames with the sampling strategy shown in Figure 3. These $n$ frames are subsequently processed by an interpolation method, denoted $g(\cdot)$, to generate a new video sequence with the original temporal length $f$. This procedure results in an interpolated dataset $\mathcal{I} = \{(g(\tilde{x}_i), \tilde{y}_i)\}_{i=1}^{|\mathcal{T}|}$, where $\tilde{x}_i \in \mathbb{R}^{n \times c \times h \times w}$ represents the set of $n$ sampled frames for the $i$-th video. A model is then trained on the dataset $\mathcal{I}$ and its performance is evaluated on

the original dataset $\mathcal{T}$. The results of this evaluation are depicted in Figure 4. With videos interpolated from single frames, the model could already achieve 88% performance of the model trained on the entire 16-frame video dataset. However, the number of pixels in the dataset $\mathcal{I}$ is only about 6% of the original dataset. These findings indicate that even when a video is reconstructed from a single randomly sampled frame, a substantial portion of the information essential to the discriminative objective can be captured by the model, while drastically reducing the pixel data processed.

This observation is essential in dataset distillation, as the majority of the dataset distillation methods (Cazenavette et al., 2022; Zhao & Bilen, 2023; Cui et al., 2023; Guo et al., 2023; Wang et al., 2024) directly updates the synthetic dataset pixel-wisely, and reducing the number of pixels is equal to reducing the number of learnable parameters during distillation process.

### 3.3 SINGLE-FRAME VIDEO SET DISTILLATION

As discussed in Section 3.2, traditional dataset distillation approaches that synthesize complete video sequences suffer from an exceptionally large parameter space, making the optimization particularly challenging. An alternative solution involves leveraging single static frames. Prior work (Wang et al., 2024) attempted to match training gradients using static frames but yielded suboptimal outcomes. We posit that a fundamental limitation of such static-frame methods is an objective mismatch. Optimizing a synthetic dataset composed of static images $\mathcal{S}_I$ to derive optimal parameters $\theta^*_{\mathcal{S}_I}$ for an image-centric model $\hat{\phi}_\theta$, does not inherently guarantee that these frames will be optimal for training a target

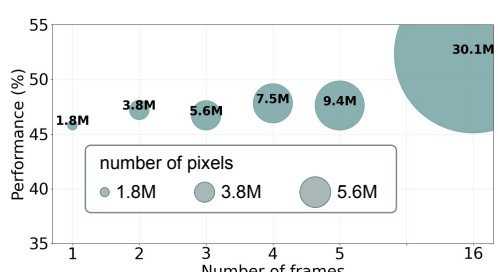

Figure 4: **Evaluation results on interpolated datasets.** With substantially fewer pixels, single-frame interpolated data retains major discrimitive information comparable to full video datasets.

video model $\phi_\theta$ to achieve its optimal parameters on the original video dataset $\mathcal{T}$. As illustrated conceptually in Figure 5 (a), the model architectures and the nature of the data (static vs. dynamic) differ, creating an inconsistency. This discrepancy means that directly optimizing static frames fails to ensure their effective generalization when used to train video models.

Instead of matching between single images, we propose Single-Frame Video set Distillation (SFVD) to match videos with images. This is directly inspired by preliminary observations in Section 3.2, which demonstrate that substantial video classification performance can be preserved when models are trained on video sequences generated via differentiable interpolation $g(\cdot)$ from a single frame, which is $l(\mathcal{T}; \theta^*_\mathcal{T}) \approx l(\mathcal{T}; \theta^*_\mathcal{I})$. This finding suggests that a single frame is capable for conveying essential discriminative information. SFVD operationalizes this insight. The synthetic dataset in SFVD, denoted $\mathcal{S} = \{(g(\hat{x}_i), \hat{y}_i)\}_{i=1}^{|\mathcal{S}|}$, comprises learnable frames $\hat{x}_i \in \mathbb{R}^{c,h,w}$ and differentiable interpolation method $g(\cdot)$. The optimization objective aims to align the training dynamics induced by these interpolated synthetic videos with those of a target dataset. In this way, the Equation 1 becomes

$$\mathcal{S}^* = \arg \min_{\mathcal{S}} \ \mathbb{E}_{\theta^{(0)} \sim \Theta}[l\left(\mathcal{I}; \theta^*_\mathcal{S}\right)]. \tag{3}$$

Using $\mathcal{I}$ as a computationally more tractable target for trajectory matching, we adapt the trajectory matching objective from Equation 2 as follows:

$$\mathcal{S}^* = \arg \min_{\mathcal{S}} \mathbb{E}_{\theta^{(0)} \sim \Theta} \sum_{t=0}^{T} \mathcal{D}\left(\theta_\mathcal{S}^{(t)}, \theta_\mathcal{I}^{(t)}\right). \tag{4}$$

Explicitly, the objective becomes:

$$\mathcal{S}^* = \arg \min_{\mathcal{S}} \mathbb{E}_{\theta^{(0)} \sim \Theta} \sum_{t=0}^{T} \mathcal{D}\left(\theta^{(t)}_{\{(g(\hat{x}_i), \hat{y}_i)\}_{i=1}^{|\mathcal{S}|}}, \theta^{(t)}_{\{(g(\tilde{x}_i), \tilde{y}_i)\}_{i=1}^{|\mathcal{T}|}}\right). \tag{5}$$

The above formulation means that by optimizing synthetic frames with $\mathcal{I}$ using the same interpolation method, SFVD ensures that the synthetic data is directly optimized for its intended use in training

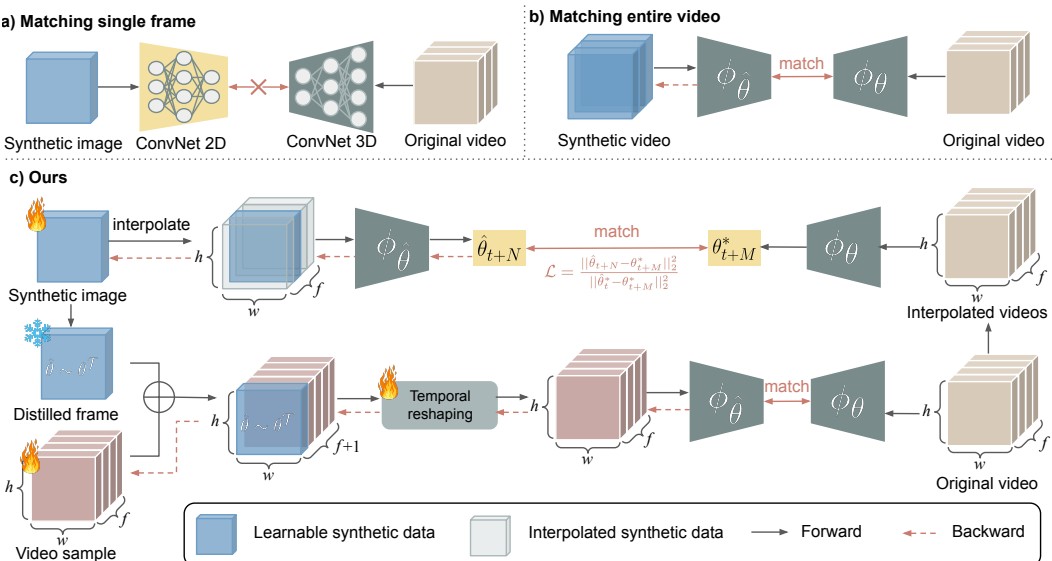

Figure 5: **Comparison of Video Dataset Distillation Strategies.** (a) Matching static single frames can lead to performance degradation due to the inherent objective mismatch between optimizing for static images and evaluating on video models. (b) Distilling entire synthetic videos, while conceptually direct, faces significant optimization challenges owing to the large parameter space involved of the synthetic dataset, as discussed in Section 3.2. (c) The proposed Single-Frame Video set Distillation (SFVD) framework optimizes a set of frames, which are subsequently interpolated into videos during the distillation process. For integration of temporal information (SFVD-T), a temporal reshaping network (TRN) is then employed to transform the intergrate the distilled frames and video samples, followed by trajectory matching with the original videos.

video models, overcoming the limitations of the method (Wang et al., 2024) focusing solely on frame matching. Additionally, SFVD significantly reduces the parameter space compared to traditional distillation methods, thus facilitating stable convergence. In summary, SFVD provides an efficient and effective approach to video dataset distillation by leveraging differentiable interpolation to bridge the representational gap between the single frame and the video data, thereby mitigating the aforementioned challenge in video set distillation.

## 3.4 INTEGRATING TEMPORAL INFORMATION

While distilled static frames can effectively capture significant semantic information from videos, the temporal information is also crucial for comprehensive video understanding. Full video sequences contain rich temporal cues but may be computationally expensive or contain redundancies. Therefore, a critical challenge arises: how to synergistically combine the information density of distilled static representations with the essential temporal context derived from video sequences? This section introduces a methodology to achieve this integration.

Let $\mathcal{S} = \{g(\hat{x}_i), \hat{y}_i\}_{i=1}^{|\mathcal{S}|}$ denote the distilled dataset by SFVD through matching with the interpolated dataset $\mathcal{I}$. We further incorporate temporal dynamics using the workflow as shown in Figure 5 (c). Specifically, we sampled $|\mathcal{S}|$ videos from the original dataset, and for each instance, both $g(\hat{x}_i)$ and the corresponding video $v_i$ are provided as input to a Temporal Reshaping Network (TRN) to obtained the fused representation $\{z_i\}_{i=1}^{|\mathcal{S}|}$, where $z_i \in \mathbb{R}^{f,c,h,w}$. The TRN fuses the distilled frame $g(\hat{x}_i)$ and its corresponding video $v_i$ by channel-wise concatenation, followed by encoding:

$$z_i = \mathcal{M}(g(\hat{x}_i) \oplus_c v_i), \tag{6}$$

where $\oplus_c$ denotes concatenation along the channel dimension.

We adopt a trajectory matching objective. Let $\theta_{t+M}^*$ denote the parameters of a target model after $M$ optimization steps on the original video dataset, starting from an initial parameter state $\theta_t^*$. Let $\hat{\theta}_{t+N}$ denote the parameters of a model trained for $N$ steps using the fused representations $z_i$, starting from the same initial state $\hat{\theta}_t = \theta_t^*$. The objective is to minimize the discrepancy between these parameter

Table 1: **Comparison with SOTA methods on various datasets.** Top-1 accuracy is reported for MiniUCF and HMDB51 dataset. Top-5 accuracy is reported for kinetics-400 and SSv2 dataset. Underlined values indicate the second best results, while **bold** values represent the overall best result.

| | Dataset | MiniUCF | | HMDB51 | | Kinetics-400 | | SSv2 | |
|---|---|---|---|---|---|---|---|---|---|
| | IPC | 1 | 5 | 1 | 5 | 1 | 5 | 1 | 5 |
| | Full Dataset | 57.2±0.1 | | 28.6±0.7 | | 34.6±0.5 | | 29.0±0.6 | |
| Coreset Selection | Random | 9.9±0.8 | 22.9±1.1 | 4.6±0.5 | 6.6±0.7 | 3.0±0.1 | 5.6±0.0 | 3.3±0.1 | 3.9±0.1 |
| | Herding (Welling, 2009) | 12.7±1.6 | 25.8±0.3 | 3.8±0.2 | 8.5±0.4 | - | - | - | - |
| | K-Center (Sener & Savarese, 2017) | 11.5±0.7 | 23.0±1.3 | 3.1±0.1 | 5.2±0.3 | - | - | - | - |
| Dataset Distillation | DM (Zhao & Bilen, 2023) | 15.3±1.1 | 25.7±0.2 | 6.1±0.2 | 8.0±0.2 | 6.3±0.0 | 9.1±0.9 | 3.6±0.0 | 4.1±0.0 |
| | MTT (Cazenavette et al., 2022) | 19.0±0.1 | 28.4±0.7 | 6.6±0.5 | 8.4±0.6 | 3.8±0.2 | 9.1±0.3 | 3.9±0.1 | 6.3±0.3 |
| | FRePo (Zhou et al., 2022) | 20.3±0.5 | 30.2±1.7 | 7.2±0.8 | 9.6±0.7 | - | - | - | - |
| | DATM (Guo et al., 2023) | 14.6±0.3 | 24.9±1.1 | - | - | - | - | - | - |
| | Static-VDSD (Wang et al., 2024) | 13.7±1.1 | 24.7±0.5 | 5.1±0.9 | 7.8±0.4 | 4.6±0.2 | 6.6±0.2 | 3.9±0.1 | 4.1±0.0 |
| | DM+VDSD (Wang et al., 2024) | 17.5±0.1 | 27.2±0.4 | 6.0±0.4 | 8.2±0.1 | 6.3±0.2 | 7.0±0.1 | 4.0±0.1 | 3.8±0.1 |
| | MTT+VDSD (Wang et al., 2024) | 23.3±0.6 | 28.3±0.0 | 6.5±0.1 | 8.9±0.6 | 6.3±0.1 | 11.5±0.5 | 5.5±0.1 | 8.3±0.2 |
| | FRePo+VDSD (Wang et al., 2024) | 22.0±1.0 | 31.2±0.7 | 8.6±0.5 | 10.3±0.6 | - | - | - | - |
| | **SFVD (ours)** | 27.5±0.7 | 34.2±0.3 | 9.7±0.3 | 13.4±1.0 | 10.4±0.4 | 15.4±0.7 | 8.0±0.5 | 10.9±0.3 |
| | **SFVD-T** | **27.8±0.2** | **36.5±1.2** | **10.9±0.6** | **15.6±0.5** | **11.4±0.2** | **16.2±0.6** | **8.3±1.1** | **11.7±0.1** |

trajectories, formalized by the loss function:

$$\mathcal{L} = \frac{||\hat{\theta}_{t+N} - \theta^*_{t+M}||^2_2}{||\hat{\theta}^*_t - \theta^*_{t+M}||^2_2}, \tag{7}$$

where the parameters $\hat{\theta}$ are updated iteratively via gradient descent for $n = 0, 1, \cdots, N-1$:

$$\hat{\theta}_{t+n+1} = \hat{\theta}_{t+n} - \alpha \nabla l(z_i, \hat{y}_i); \hat{\theta}_{t+n}), \tag{8}$$

where $l(\cdot)$ represents the cross-entropy loss, and $\alpha$ is the learning rate. Crucially, during this optimization phase, the distilled representations $\hat{x}_i$ are treated as fixed inputs. In contrast, the parameters within the TRM $\mathcal{M}$ and potentially any parameters involved in the processing of the video segments $v_i$ are learnable. This allows the temporal integration mechanism to adapt and optimize the fusion process to effectively mimic the training trajectories on the original dataset.

## 4 EXPERIMENTS

### 4.1 DATASETS AND METRICS

Our evaluation methodology aligns with prior research (Wang et al., 2024), employing the UCF101 (Soomro et al., 2012), HMDB51 (Kuehne et al., 2011), Kinetics-400 (Carreira & Zisserman, 2017), and Something-Something V2 (Goyal et al., 2017) datasets. UCF101 contains 13,320 videos distributed across 101 action categories, whereas HMDB51 includes 6,849 videos categorized into 51 classes. Kinetics (Carreira & Zisserman, 2017) is a comprehensive collection of video clips spanning 400 / 600 / 700 human action classes. Something-Something v. 2 (SSv2) (Goyal et al., 2017) focuses on 174 motion-intensive classes. For the UCF101 and HMDB51 datasets, we report top-1 classification accuracy. In line with (Wang et al., 2024), we also utilize MiniUCF, a subset of UCF101 that includes its 50 most prevalent classes. For the SSv2 and Kinetics-400 datasets, our reported metric is top-5 classification accuracy.

### 4.2 IMPLEMENTATION DETAILS

Following previous work (Wang et al., 2024), the videos of MiniUCF and HMDB51 are sampled to 16 frames, with a sampling interval of 4 frames dynamically. Each frame is cropped and resized to 112×112. For Kinetics-400 and SSv2, the video are sampled to 8 frames and the frames are cropped to 64×64. For SFVD interpolation, the synthetic frames are duplicated 16 times on MiniUCF and HMDB51, 8 times on Kinetics-400 and Something-Something V2. When the number of required frames is greater than one, the subsequent interpolation was performed consistently with the procedures outlined in Figure 3. Since HMDB51, SSv2, and Kinetics are highly class-imbalanced datasets, we oversample the less frequent samples within these datasets (Zhao et al., 2024). Hyperparameters are detailed in Appendix A.

### 4.3 MAIN RESULTS

**Compare with SOTA methods.** We compared our method with state-of-the-art image and video distillation techniques, as well as coreset selection methods, on four widely used video datasets:

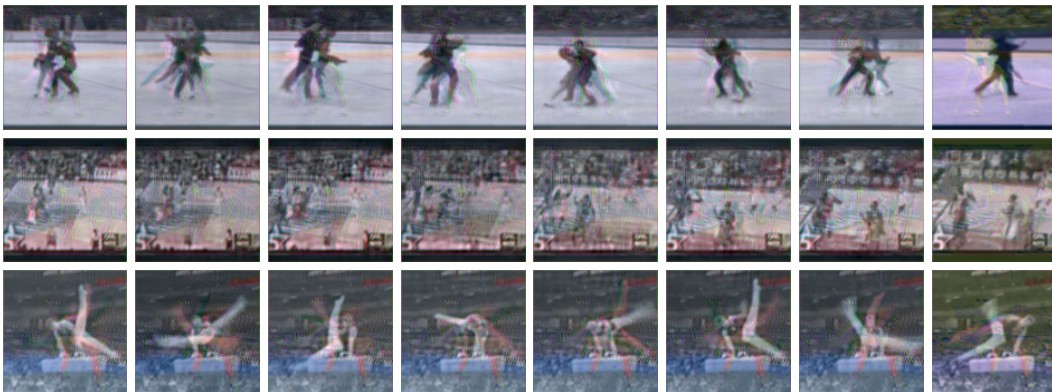

Figure 6: **Visualization of the distilled dataset.** We visualized 3 samples of the distilled datasets of MiniUCF with IPC=5. For simplicity we only show 8 frames of the samples. The exhibited samples from up to down are from "IceDancing", "Basketball", and "PommelHorse".

MiniUCF, HMDB51, Kinetics-400, and Something-Something V2. As shown in Table 1, our SFVD method outperforms all existing approaches on the MiniUCF dataset, even without using temporal information. This is attributed to our method's more constrained parameter updating space, which helps capture discriminative features more effectively. SFVD is also highly storage-efficient, requiring only one image per instance (IPC).Furthermore, the integration of temporal information (SFVD-T) leads to an even better performance of the distilled dataset. We found that SFVD performs especially well with smaller IPC values, and similar performance characteristics and advantages of our methods were also observed on the HMDB51 dataset.

For larger video datasets such as Kinetics-400 and Something-Something V2, our proposed method also demonstrates leading performance. These more extensive and complex datasets often pose significant challenges for distillation techniques due to their increased diversity in actions, scenes, and temporal dynamics. Despite these challenges, our approach effectively identifies and preserves crucial semantic information, resulting in a distilled dataset that maintains a high level of accuracy. This indicates the robustness and scalability of our method, showcasing its capability to handle the increased complexity and data volume inherent in larger-scale video analysis tasks, while still achieving superior results compared to the existing methods.

**Cross architecture evaluation.** Besides ConvNet3D, CNN+GRU (Donahue et al., 2015) and CNN+LSTM (Yue-Hei Ng et al., 2015) model architectures are commonly used for video recognition. Therefore, following the setting of the previous work (Wang et al., 2024), we conduct the cross-architecture experiments as shown in Table 2. As indicated in the table, our method also shows a better architecture generalization of the distilled dataset.

Table 2: **Cross-architecture evaluation.** Experiments are conducted on the MiniUCF dataset with one image per class (IPC = 1).

| Method | Evaluation Model | | |
|---|---|---|---|
| | ConvNet3D | CNN+GRU | CNN+LSTM |
| Random | 9.9±0.8 | 6.2±0.8 | 6.5±0.3 |
| DM | 15.3±1.1 | 9.9±0.7 | 9.2±0.3 |
| MTT | 19.0±0.1 | 8.4±0.5 | 7.3±0.4 |
| DM+VDSD | 17.5±0.1 | 12.0±0.7 | 10.3±0.2 |
| MTT+VDSD | 23.3±0.6 | 14.8±0.1 | 13.4±0.2 |
| **ours** | **27.8±0.2** | **19.3±0.5** | **18.4±0.4** |

### 4.4 ABLATION STUDIES

**Ablation Analysis of Proposed Components.**
To verify the individual contribution of each component in our method, we conducted a comprehensive ablation study. The empirical results, detailed in Table 3, demonstrate that using the SFVD module alone significantly improves performance over the baseline (Guo et al., 2023), indicating its effectiveness in capturing essential visual patterns even without temporal cues. Furthermore, when the integration of temporal information is involved, we observe a further enhancement in performance, suggesting that temporal dynamics play a complementary role in enhancing representation quality. The consistent gains across multiple evaluation metrics confirm that both components contribute meaningfully to the effectiveness of our approach.

**Impact of Varying the Number of Distilled Frames.** In an extension of our investigation into the distillation process, we explored alternative configurations to the single-frame distillation approach.

Table 3: **Component ablation.**

| Method | IPC=1 | IPC=5 |
|--------|-------|-------|
| DATM | 14.6±0.3 | 24.9±1.1 |
| SFVD | 27.5±0.7 | 34.2±0.3 |
| SFVD-T | **27.8±0.2** | **36.5±1.2** |

Table 4: **# frames selection.**

| # frames | IPC=1 | IPC=5 |
|----------|-------|-------|
| 1 | **27.5±0.7** | **34.2±0.3** |
| 2 | 26.8±0.6 | 32.8±0.4 |
| 3 | 23.4±1.8 | 30.3±0.6 |

Table 5: **Soft-label influence.**

| Softlabel | IPC=5 | Δ Acc. |
|-----------|-------|--------|
| VDSD-S | 14.4±1.9 | 13.9↓ |
| SFVD-S | 34.2±0.8 | 4.1↑ |
| SFVD-TS | 26.5±0.8 | 10.0↓ |

Specifically, experiments were designed to evaluate the effects of distilling video sequences into a varying number of representative frames (1, 2, and 3). The subsequent interpolation was performed consistently with the procedures outlined in Figure 3. The quantitative evaluation outcomes are presented in Table 4. As the number of frames targeted for distillation increases, there is a corresponding degradation in overall performance. This empirical finding aligns with and substantiates the concerns articulated in Section 3. Specifically, an increase in the number of distilled frames leads to a significant expansion in the parameter space of the distillation model. This expansion can complicate the optimization landscape, potentially preventing the distillation process from achieving robust convergence and leading to suboptimal performance.

**Investigation of Soft Label Integration.** Recognizing the significant emphasis placed on the utility of soft labels within recent dataset distillation works (Guo et al., 2023; Sun et al., 2024; Yin et al., 2023; Qin et al., 2024; Su et al., 2024; Shao et al., 2024), we conducted a series of experiments to evaluate their integration with our proposed methodology. For this investigation, the soft labels employed were derived directly from model logits, rather than utilizing more elaborate soft labels generated through complex knowledge distillation (KD) techniques (Sun et al., 2024; Yin et al., 2023; Shao et al., 2024).

The results are presented in Table 5. Notably, our proposed SFVD framework demonstrated effective compatibility with soft labels, maintaining or enhancing its performance. In contrast, both the baseline VDSD method (Wang et al., 2024) and our temporally-aware SFVD-T variant exhibited a decline in performance when soft labels were introduced. This divergence in outcomes is hypothesized to stem from the increased complexity introduced by a larger number of learnable parameters during the distillation phase, particularly for VDSD and SFVD-T in conjunction with soft labels. An over-parameterized distillation process can lead to optimization challenges and instability. Consequently, to ensure robust convergence and facilitate the reliable determination of optimal parameters for temporal reshaping and video sample selection, hard labels were selected for the primary experiments, prioritizing stability and consistent results in those contexts.

### 4.5 QUALITATIVE RESULTS

To observe the temporal changes in the distilled videos, we sampled frames from the videos obtained using different methods and visualized their inter-frame differences. We show three samples in Figure 6 and more in the Appendix C. Although visually abstract, we can still conclude that the distilled videos contain information from more than a single video and indeed exhibit temporal variations. These variations suggest that the distilled videos are not static repetitions but carry meaningful transitions over time, capturing diverse motion patterns that contribute to effective video model training. This validates the effectiveness of the proposed method in preserving temporal information within the distilled dataset.

### 5 CONCLUSION

In this paper, we confronted the critical challenge of a drastically increased number of parameters in video dataset distillation. We introduced the novel Single-Frame Video set Distillation (SFVD) framework, premised on the insight that individual frames often contain substantial discriminative information. SFVD initially distills highly representative frames, reducing synthetic data dimensionality and simplifying optimization. Subsequently, it integrates essential temporal information using a Temporal Reshaping Network (TRN) that fuses these distilled frames with video samples, and then matches with the original video features. Extensive experiments confirmed that SFVD significantly outperforms existing methods, validating our strategy of first tackling dimensionality and then incorporating temporal information. This research underscores a successful approach to the unique challenges of video data, offering a path towards more efficient and effective distilled video datasets and scalable training for video recognition models, while opening avenues for future exploration in advanced temporal modeling and broader task applications.

## ETHICS STATEMENT

We have carefully reviewed and adhered to the ICLR Code of Ethics. Our work focuses on video dataset distillation and relies solely on publicly available benchmark datasets (UCF101, HMDB51, Kinetics-400, and Something-Something V2). These datasets are widely used in the research community, and we follow their respective licenses and intended uses. No private, sensitive, or personally identifiable information is included. We also acknowledge possible broader impacts, including risks of information loss, security vulnerabilities, or intellectual property concerns, as noted in Appendix B. We encourage responsible use of our method and datasets in alignment with ethical AI development practices.

## REPRODUCIBILITY STATEMENT

We have made extensive efforts to ensure reproducibility of our results. Details of dataset preprocessing, model architectures, training schedules, and hyperparameters are provided in Section 4 and Appendix A of the paper. All datasets used are publicly accessible, and our evaluation protocols are consistent with prior work to ensure comparability. We will release our implementation and instructions for reproducing all experiments in an anonymized repository with the submission, and will make the code and supplementary materials publicly available upon acceptance.

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

## A  TRAINING DETAILS

We report the most important hyperparameters in the SFVD distillation in Table 6, where `lr_img` stands for the learning rate to update the synthetic frames, `lr_y` denotes the learning rate for the soft-label optimizer, `lr_lr` represents the learning rate for adaptive lr, `batch_syn` indicates the number of synthetic frames to match at each iteration, and `syn_steps` is the steps of training trajectories of the synthetic dataset to match expert trajectories.

Table 6: Hyperparameters for SFVD on video datasets.

| Dataset | IPC | lr_img | lr_y | lr_lr | batch_syn | syn_steps |
|---|---|---|---|---|---|---|
| miniUCF | 1 | 1000 | 10 | $1 \times 10^{-5}$ | 50 | 40 |
|         | 5 | 1000 | 10 | $1 \times 10^{-5}$ | 50 | 80 |
| HMDB51 | 1 | 1000 | 10 | $1 \times 10^{-5}$ | 51 | 40 |
|        | 5 | 1000 | 10 | $1 \times 10^{-5}$ | 51 | 80 |
| Kinetics | 1 | 1000 | 10 | $1 \times 10^{-5}$ | 32 | 40 |
|          | 5 | 1000 | 10 | $1 \times 10^{-5}$ | 32 | 40 |
| SSv2 | 1 | 1000 | 10 | $1 \times 10^{-5}$ | 32 | 40 |
|      | 5 | 1000 | 10 | $1 \times 10^{-5}$ | 32 | 40 |

## B  LIMITATIONS AND BROADER IMPACTS

Video dataset distillation is currently in its early stages. While image dataset distillation methods have achieved nearly lossless performance on small datasets (Guo et al., 2023), our approach to video set distillation still shows a performance gap compared to using the full dataset.

Our work presents a mixed bag of societal impacts. On the positive side, it enhances resource efficiency, accelerates AI development, improves accessibility, and can strengthen data privacy. On the other hand, it carries risks such as losing information, introducing security vulnerabilities, and complicating intellectual property rights. Balancing these benefits and challenges is essential to harness the technology's potential responsibly and ethically.

## C  MORE QUALITATIVE RESULTS

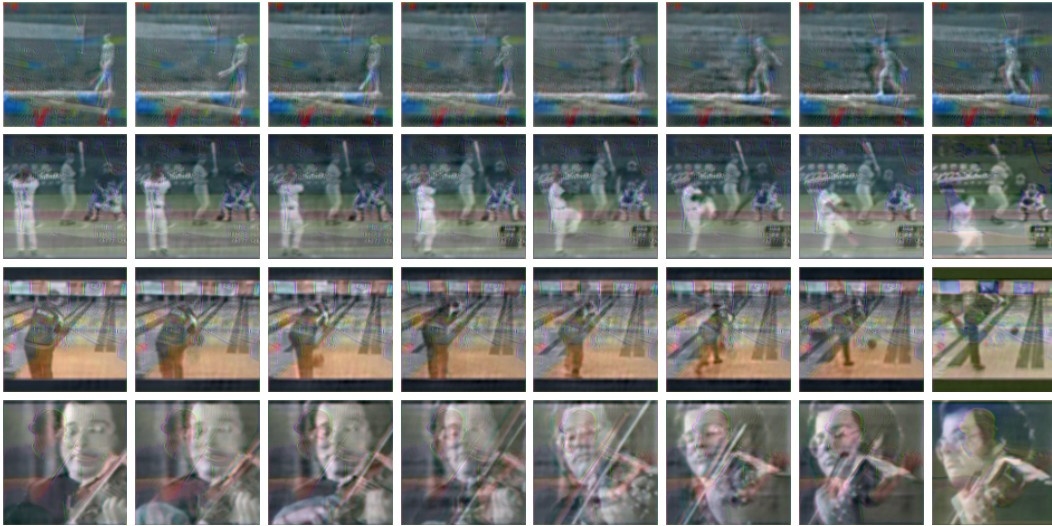

Figure 7: Qualitative results of the distilled dataset. From top to bottom, the classes depicted are "BalanceBeam", "BaseballPitch", "Bowling", and "PlayingViolin."

## D  USE OF LARGE LANGUAGE MODELS (LLMs)

In preparing this paper, we made limited use of large language models (LLMs) as general-purpose assistive tools. Specifically, LLMs were employed for (i) improving the clarity and conciseness of English writing, including grammar correction and refinement of paragraph structure, and (ii) generating alternative phrasings to enhance readability. Importantly, all technical ideas, experimental designs, theoretical insights, and substantive claims presented in this work were conceived, implemented, and validated entirely by the authors.

LLMs were not used to generate novel research ideas, conduct experiments, or produce results. The authors take full responsibility for the content of this submission, including any sections drafted with the assistance of LLMs. All content has been carefully reviewed to ensure accuracy, originality, and compliance with the ICLR Code of Ethics.

