# OpenReview forum: "Distilling Video Datasets into Images"
_ICLR.cc/2026/Conference — ICLR 2026 Conference Withdrawn Submission_

### Official Review · Reviewer_Amqo · 2025-10-30

**Soundness:** 2
**Presentation:** 2
**Contribution:** 2
**Rating:** 2
**Confidence:** 4

**Summary:**

This paper addresses the problem of video dataset distillation, highlighting the challenges that arise when extending existing image dataset distillation methods to the video domain. Specifically, the explosive increase in learnable parameters and the resulting optimization instability. The authors begin with the observation that the core semantic content of a video is often sufficiently captured within a single frame, and based on this insight, they propose a novel framework called Single-Frame Video Distillation (SFVD).

**Strengths:**

* By adopting an intuitive and straightforward approach of “distilling videos into images”, the proposed method effectively simplifies the otherwise complex problem of video distillation.
* It substantially reduces the parameter space, thereby achieving both training stability and computational efficiency.
* Furthermore, the work introduces a novel research direction that bridges the gap between image and video distillation frameworks.

**Weaknesses:**

Refers to questions.

**Questions:**

1. In Table 1, it would be helpful to clarify the accuracy performance of the target parameters for each full video dataset, so readers can better understand the relative improvement achieved by SFVD and SFVD-T.
2. Compared with SFVD, what advantages does SFVD-T offer in terms of memory storage and computational cost? It is also worth examining whether SFVD-T theoretically represents an additional knowledge-distillation stage built upon the single-frame distilled data obtained from SFVD.
3. What is the rationale for adopting the trajectory matching objective? Since other loss functions such as gradient matching or distribution matching could also be applied, how do their numerical results and qualitative outcomes compare with those of trajectory matching?
4. It would be valuable to disclose the detailed experimental environment, including the degree of data condensation (memory storage) and the GPU memory usage during training for both SFVD and SFVD-T.

---

### Official Review · Reviewer_EBp7 · 2025-10-31

**Soundness:** 3
**Presentation:** 3
**Contribution:** 2
**Rating:** 6
**Confidence:** 2

**Summary:**

The paper introduces Single-Frame Video set Distillation (SFVD), a method that compresses large video datasets into a small set of learnable key frames. Each distilled frame is interpolated into video sequences and refined using a Temporal Reshaping Network to retain temporal cues. Across diverse benchmarks, SFVD achieves consistent accuracy gains over prior video distillation methods while being far more efficient.

**Strengths:**

1. Novel and simple formulation: SFVD identify the redundant information issues in video data and reformulates video distillation as a single-frame optimization problem.
2. Strong empirical results with high efficiency: SFVD demonstrates sota performance across multiple benchmarks  with significantly lower computation and storage costs.
3. Comprehensive ablations: Detailed ablations and cross-architecture experiments support SFVD design choices and show consistent improvements over prior baselines.

**Weaknesses:**

1. Motion capture: Although the proposed TRN can recover motion patterns, the distilled representation is still mainly derived from static single frames, which may fail to capture complex temporal dependencies such as long-range interactions or causal motion dynamics. Some analysis or discussion on whether motion is effectively captured should be provided.
2. Assumption limitation: The SFVD build on the observation that video data usually have redundant information, which is reasonable. However, the assumption that one image can contain sufficient information within a video seem limited (by video duration and dynamics) and has no support. The ablation and main results are promising, however I am curious whether such method can be applied to longer video data or videos with more dramatic motions.

**Questions:**

1,2: See wealkness

3. Following last question, I consider SFVD a more semantics driven method. It presents promising results even without temporal components. I am curious whether semantics are dominant factors (compared to motions) in video data distillation and whether we need more motion heavy benchmarks for this task? Look forward to authors' insight and discussions.

---

### Official Review · Reviewer_Gf2U · 2025-11-01

**Soundness:** 3
**Presentation:** 2
**Contribution:** 3
**Rating:** 6
**Confidence:** 3

**Summary:**

This work addresses a core challenge in video dataset distillation: the massive, learnable parameter space introduced by the temporal dimension of video data. The authors argue that this explosion in parameters is a primary cause of the convergence difficulties and suboptimal performance seen in existing methods. To tackle this, this work proposes a framework called Single-Frame Video set Distillation (SFVD). The core idea is to distill each video class into a single, highly informative synthetic frame, rather than a full video sequence. During the distillation process, this single learnable frame is transformed back into a video sequence via a differentiable interpolation function, which is then used for training trajectory matching against the original dataset. By constraining the optimization to a single frame, the number of learnable parameters is drastically reduced, leading to a more efficient and stable optimization process. Furthermore, to incorporate the temporal information that a single frame might miss, this work proposes  SFVD-T, which employs a Temporal Reshaping Network (TRN) to fuse the distilled frame with a real video clip sampled from the original dataset.
The authors conduct extensive experiments on several standard video benchmarks, including MiniUCF, HMDB51, Kinetics-400, and Something-Something V2. The results demonstrate that the proposed method significantly outperforms previous works across various settings.

**Strengths:**

1. The core idea of distilling video data into a single frame is interesting and well-designed. By restricting the optimization space to the "image" domain while keeping the matching process in the "video" domain, it decouples the complexity of parameter learning from the domain requirements of the task. Using a differentiable interpolation function as a bridge between these two domains is a simple yet highly effective solution.

2. This work evaluates its method on four video datasets against a diverse set of baselines, including coreset selection, image distillation, and video distillation methods. The results in Table 1 comprehensively demonstrate the effectiveness of the proposed approach.

3. The experiment on the number of distilled frames (Table 4) shows that performance degrades as more frames are added to the distilled set. This directly validates the paper's central hypothesis that a larger parameter space hinders optimization and is a powerful piece of evidence supporting the method's design.

**Weaknesses:**

1. The Differentiable Interpolation Function is the critical link between the distilled image and the matched video. The implementation details (Section 4.2) state that for SFVD, this operation is to "duplicate 16 times." If it is simple frame duplication, the term "differentiable interpolation" might be an overstatement, as no true interpolation occurs. The authors need to clarify the exact design of this function and discuss whether more sophisticated interpolation methods were considered and what their impact might be.

2. The SFVD-T variant requires sampling real videos from the original dataset during distillation. This seems to conflict with the core goal of dataset distillation, which is to create a small, standalone synthetic dataset that entirely replaces the original. Does this imply that even after distillation with SFVD-T, the original dataset must be retained for future use (e.g., fusing data when training a new model on the distilled set)?

**Questions:**

1.  Could you please elaborate on the exact form of the "differentiable interpolation function" used in your experiments? Beyond simple frame duplication, did you experiment with other, more complex interpolation strategies? How sensitive is the model's performance to the choice of the  interpolation function?

2. In SFVD-T, the TRN fuses a distilled representation $g(\hat{x}_i)$ with its "corresponding video $v_i$." Since $\hat{x}_i$ is a learned synthetic representation for a class, it has no direct one-to-one correspondence with a specific original video $v_i$. How is this $v_i$ selected? Is it a randomly sampled video from the same class?

3. The results in Table 5 show that soft labels improve performance for SFVD but are detrimental to SFVD-T and the baseline VDSD. The authors attribute this to the increased complexity of the learnable parameters. Does this suggest that more complex distillation models (like SFVD-T) might require stronger regularization or different optimization strategies to effectively leverage the information provided by soft labels?

---

### Official Review · Reviewer_2z4x · 2025-11-03

**Soundness:** 2
**Presentation:** 2
**Contribution:** 2
**Rating:** 2
**Confidence:** 4

**Summary:**

This paper proposes Single-Frame Video Set Distillation (SFVD), a method for condensing large-scale video datasets into a small number of representative frames. The core idea is that a single frame can capture most of a video’s discriminative semantics, thereby reducing optimization difficulty caused by the large parameter space in video distillation. The method further introduces a Temporal Reshaping Network (TRN) to fuse temporal information from real videos and distilled frames.

**Strengths:**

1. Novel perspective on reducing video distillation complexity by exploiting the semantic sufficiency of single frames.
2. Comprehensive experimental evaluation across four video benchmarks.
3. The integration of TRN effectively bridges static frame distillation and temporal dynamics, yielding consistent gains.

**Weaknesses:**

1. The paper contains redundant explanations and repeated arguments (e.g., that a single frame captures complete video semantics), which affect readability. The narrative could be more concise and better structured.

2. Unclear core component design:
  - The method for generating interpolated videos from single frames (g(·)) is not concretely described;
  - The structure and role of $\mathcal{M}$ in Eq. (6) within TRN are insufficiently explained;

3. It is not explained how the TRN branch (Sec. 3.4) enhances the synthetic frames. As shown in Fig. 5, gradients are not backpropagated to the synthetic images.

4. TRN training uses both real videos and synthetic frames jointly, which may introduce potential data leakage and unfair comparison with methods trained solely on synthetic data. The authors should clarify whether real data are involved at the time of inference or evaluation.

**Questions:**

1. How is the differentiable interpolation function g(·) implemented?
2. Does $\mathcal{M}$ in TRN share parameters with the downstream model, or is it an independent module?
3. During training, does the TRN branch influence the synthetic frames indirectly, or are the frames completely frozen once generated?
4. Would the proposed SFVD-T still perform well if the TRN were trained without access to real videos?

---

### Note · Authors · 2025-11-13

I have read and agree with the venue's withdrawal policy on behalf of myself and my co-authors.